# Early Feeding Regime of Waste Milk, Milk, and Milk Replacer for Calves Has Different Effects on Rumen Fermentation and the Bacterial Community

**DOI:** 10.3390/ani9070443

**Published:** 2019-07-15

**Authors:** Rong Zhang, Wei-bing Zhang, Yan-liang Bi, Yan Tu, Yves Beckers, Han-chang Du, Qi-yu Diao

**Affiliations:** 1Feed Research Institute, Chinese Academy of Agricultural Sciences, Beijing Key Laboratory for Dairy Cow Nutrition, Beijing 100081, China; 2Precision Livestock and Nutrition Unit, Gembloux Agro-Bio Tech, University of Liège, Passage des Déportés, 2, 5030 Gembloux, Belgium; 3Shandong Agricultural Biological Immune Technology Engineering Laboratory, Shandong Yinxiang Weiye Group Co., Ltd, Heze 274400, China

**Keywords:** waste milk, whole milk, milk replacer, rumen development, rumen microbiota, ruminal imprinting

## Abstract

**Simple Summary:**

The postnatal period may be the most critical window for rumen manipulation, and the early feeding regime may lead to permanent changes in the rumen microbial composition. The objective of this research was to investigate the effects of the most common liquid feeds (whole milk, waste milk, and milk replacer) on growth performance, rumen development, and the ruminal bacterial community during the weaning period, and to then follow calves to up to six months of age to determine the persistence of any ruminal imprinting effects. The results demonstrate that the early feeding regime impacts rumen development not only by dry matter intake, but also the type of liquid feed. Calves fed waste milk had a distinctly structured bacterial community at two months of age, but this difference diminished at six months of age. Calves fed milk replacer had a different rumen fermentation pattern at two months of age, which may induce a long-lasting effect on the rumen environment.

**Abstract:**

We investigated the effects of different types of early feeding on rumen fermentation parameters and the bacterial community in calves. Fifty-four Holstein calves were assigned to three treatments and fed whole milk (M), pasteurized waste milk (WM), or milk replacer (MR). Male calves were slaughtered at the age of two months to measure the stomach masses. The female calves were followed for six months to determine the body weight, blood indices, rumen fermentation, and ruminal bacterial community. At the age of two months, the average daily gain was lower, but the concentration of total volatile fatty acids was greater in the MR group. Starter intake and stomach mass were lower, but the isovalerate molar proportion was greater in the WM group. The blood indices and ruminal bacterial community of the WM group differed from those of the other groups. At the age of six months, the ruminal propionate molar proportion was lower, but the ruminal pH and acetate/propionate ratio were greater in the MR group. In conclusion, calves fed WM had different rumen fermentation and bacterial community during the weaning period, whereas feeding MR produced a long-lasting effect on the rumen environment.

## 1. Introduction

Whole milk (M), waste milk (WM), and milk replacer (MR) are the most common liquid feeds for calves on dairy farms. WM is comprised of colostrum, milk obtained from mastitic cows, and milk from cows treated with antibiotics. The growth performance has been shown to be similar for calves fed WM and M [1,2], whereas the growth rate of calves fed MR is influenced by the ingredient composition and nutrient intake of the MR [3,4]. As most liquid feed flows directly into the abomasum upon suckling action, which closes the esophageal groove, the intestine, rather than the rumen, is the major digestion site for pre-weaned calves. Therefore, previous research has mainly focused on the effects of different liquid feeds on intestinal microbiota and development. Colostrum was shown to have a positive effect on gastrointestinal tract development and function in calves, not only through the provision of nutrients, but also due to the high concentration of growth factors and biologically active peptides [5,6]. Calves fed pasteurized WM had a more diverse bacterial community in feces [7], whereas calves fed MR containing soy flour had a less acidic abomasal environment [8,9] and slower intestinal development [10]. 

As the rumen develops and is colonized by microorganisms, a calf physiologically transits from a non-ruminant to ruminant state. The postnatal period may be the most critical window for rumen manipulation, and the early feeding regime may lead to long-lasting changes in the rumen microbial composition [11,12,13]. Diets could modify the establishment of the bacterial community of lambs around the time of weaning, and this modification persists over four months [14]. De Barbieri (2015) [15] also found that ruminal bacterial communities of lambs can be altered by the diet of the maternal ewes and lambs or by inoculums from donor ewes fed with different diets, and the difference lasts until five months of age. However, not all nutritional interventions in the early life of calves can promote the establishment of different microbial populations in the rumen of the young animals. Dietary supplementation with sanguinarine and resveratrol did not yield long-term effects on rumen fermentation patterns or the bacterial community in calves [16]. Therefore, the aim of the current study was to evaluate the effects of the most common liquid feeds on calf ruminal development, fermentation, and the bacterial community during the weaning period, and to then follow them up to six months of age to determine the persistence of any effects.

## 2. Materials and Methods 

### 2.1. Animals, Treatments, and Management

Fifty-four Holstein calves (*n* = 15 per treatment, female; *n* = 3 per treatment, male) were recruited from the Yin Xiang Dairy Farm (Shandong, China) and the average age of calves was 2.8 days. Calves were fed 4 L of colostrum by esophageal feeder immediately after they were born. Following this, calves were accustomed to bucket feeding with M, WM, or a transition MR, respectively. The transition MR was composed of 50% M and 50% MR. From one to six days of age, the feeding amount was 5 L per day. 

The experiment began when calves were seven days of age and terminated when they were 180 days old. At seven days of age, calves were randomly assigned to one of three treatments and fed M, pasteurized WM, or MR, respectively. The MR was reconstituted as an emulsion (12.5%, w/v) in cooled (40 °C) water. Calves were bucket fed twice a day. The amount of M, WM, or MR fed to the calves was 12% v/w of body weight per day (Table 1). None of the liquid feed was left. A pelleted feed was provided ad libitum to the calves from 14 days of age onwards. Male calves (*n* = 3 per treatment) were slaughtered at 58 days of age. Female calves (*n* = 15 per treatment) were weaned gradually from 60 to 63 days of age. From 7 to 63 days of age, calves were raised in separate calf hutches (1.2 × 3 m). From 64 to 180 days of age, calves were raised together according to their treatment in naturally ventilated barns. A pelleted feed and forage were provided separately ad libitum from 64 to 180 days of age. The nutrient profiles of M, WM, and MR are listed in Table 2. The ingredient composition and nutrient profile of the concentrate feed and forage are presented in Table 3. The nutrient profile of the liquid feed, concentrate feed and forage were analyzed according to the Association of Official Analytical Chemists (AOAC 1990, Washington, DC, USA) protocols. Crude lactose was analyzed using high performance liquid chromatography (HPLC) according to procedures described previously [17]. Gentamicin was analyzed using enzyme linked immunosorbent assay (ELISA) kits [18]. The average concentration of gentamicin in WM was 67 ± 42 μg/L (mean ± SD). The experimental protocol (protocol number: AEC-CAAS-2015-01) was approved by the Animal Ethics Committee of the Chinese Academy of Agriculture Science (Beijing, China).

### 2.2. Sampling and Measurements

#### 2.2.1. Growth Performance and Feed Intake

The body weight of calves was measured before the morning feeding when the calves were 7, 14, 28, 49, 58, 90, 120, 150, and 180 days old. The offered pelleted feed and refusals were weighed daily to calculate the feed intake on the seven consecutive days prior to weaning at 60 days old (female calves, *n* = 15 per each treatment).

#### 2.2.2. Blood Metabolites and Hormone Measurements

Blood was sampled before the morning feeding at 60 and 180 days of age (female calves, *n* = 10 per treatment). The blood samples were centrifuged at 1350× *g* for 20 min at 4 °C. The serum was decanted and stored at −20 °C before being used to determine the concentration of serum urea nitrogen (SUN), non-esterified fatty acids (NEFA), growth hormone (GH), insulin (INS), insulin-like growth factor-1 (IGF-1), and human epidermal growth factor (h-EGF). The SUN and NEFA concentrations were determined using a Model 7600 automatic biochemical analyzer (Hitachi, Tokyo, Japan) [19]. The insulin concentration was determined using a radioimmunoassay kit (Beijing SINO-UK Institute of Biological Technology, Beijing, China) and a GC-911-γ-Radiation immunity arithmometer (Zhongke zhongjia Scientific Instruments Co., Ltd., Anhui, China) [20]. The concentrations of GH, IGF-1, and h-EGF were measured using ELISA kits (Beijing SINO-UK Institute of Biological Technology, Beijing, China) and a Stat Fax 2100 microplate reader (Awareness Technology Inc, Palm City, FL, USA) [21].

#### 2.2.3. Assessment of Rumen Organ Development

Male calves were slaughtered at 58 days of age (*n* = 3 per treatment). Stomach compartments were collected, emptied, cleaned with saline, drip dried, and weighed. The stomach mass of calves was calculated as a percentage of live weight, and the wet mass of each stomach compartment was calculated as a percentage of the total weight of the four stomachs.

#### 2.2.4. Rumen Fermentation Parameter Measurements

Ruminal liquid was sampled at 60 and 180 days of age from calves fed WM, M, or MR, respectively (WM2, M2, MR2, WM6, M6, MR6) (female calves, *n* = 8 per treatment). Ruminal liquid was collected 2 to 3 h after the morning feeding using an oral stomach tube. The first 100 mL of fluid was discarded, and the remainder was saved. Each sample was individually filtered through a double layer of gauze and collected in a clean tube. Samples (15 mL) placed individually into vacuum tubes were kept at −20 °C until the analysis of volatile fatty acids (VFA) and ammonia nitrogen (NH_3_-N). Other tubes containing 2 mL of the ruminal liquid samples were placed into liquid nitrogen prior to an analysis of the bacterial community. Ruminal pH values were determined immediately after sampling using a digital Basic PB-20 pH meter (Sartorius AG, Göttingen, Germany). The NH_3_-N concentration was determined using the phenol hypochlorite colorimetric method [22]. Briefly, 50 μL of rumen fluid filtrate was added to 2.5 mL phenol reagent, and then mixed with 2.0 mL hypochlorite reagent. The mixture was incubated at 95 °C in a water bath for 5 min, and the absorbance value was measured immediately at 630 nm after cooling. The VFA concentration was determined using an SP-3420 gas chromatograph system (Beijing Analytical Instrument Factory) according to Zhang and colleagues (2016) [23]. Briefly, 1mL of rumen fluid filtrate was mixed with 25% metaphosphoric acid solution which contained 2% 2-ethyl butyrate and was then frozen at −20 °C overnight. After thawing, the samples were centrifuged and the supernatants were analyzed by gas chromatography. The chromatography conditions were as follows: PEG-20M + H_3_PO_4_ column, 2 m × 6 mm × 2 mm; column temperature, 200 °C; carrier gas, nitrogen; gas flow, 30 mL/min; flame ionization detector temperature, 200 °C; injector temperature, 200 °C; and injection volume, 0.6 μL.

#### 2.2.5. Identification of the Rumen Bacterial Community

DNA Extraction, PCR Amplification, and Illumina Sequencing: Total DNA was extracted from 1 mL of each ruminal liquid sample using the QIAamp Fast DNA Stool Mini kit (QIAGEN, Germany) according to the manufacturer’s instructions. Amplification by polymerase chain reaction (PCR) was conducted with the 515f/806r primer set that amplifies the V4 region of the 16S rRNA gene (515F: 5′-GTG CCA GCM GCC GCG GTA A-3′; 806R 5′-XXX XXX GGA CTA CHV GGG TWT CTA AT-3′) [24]. The reverse primer contained a 6-bp error-correcting barcode unique to each sample. PCR amplifications were performed in a 30 μL mixture containing 15 μL of 2 × Phusion High-Fidelity PCR Master Mix (New England Biolabs, USA), 2 μM of forward and reverse primers, 10 μL of a 1 ng/μL DNA template, and 2 μL of high-performance liquid chromatography (HPLC)-grade water. The cycling conditions consisted of an initial cycle of 98 °C for 1 min, followed by 30 cycles of 98 °C for 10 s, 50 °C for 30 s, and 72 °C for 30 s, and a final cycle of 72 °C for 5 min. The PCR products were excised from a 2% (w/v) agarose gel and purified using a GeneJET Gel Extraction kit (Thermo Scientific). Illumina paired-end sequencing libraries were constructed using the NEBNext DNA sample preparation kit (New England Biolabs, USA). DNA quality was checked using the Agilent 2100 Bioanalyzer (Agilent Technologies, Palo Alto, CA) followed by quantification on the Qubit 2.0 Fluorometer (Life Technologies, Carlsbad, CA, USA). Sequencing of amplified bacterial 16S rRNA gene fragments was performed using an IlluminaHiSeq2500 platform (Novogene Bioinformatics Technology Co., Ltd., Beijing, China). 

#### 2.2.6. Bioinformatic Analysis

Pairs of reads from the original DNA fragments were assigned to each sample according to the unique barcodes and were merged using FLASH (v1.2.7) [25]. Raw tags were quality filtered using QIIME (v1.7.0) [26]. Chimeric sequences were identified by a comparison with those in the Gold database (version microbiomeutil-r20110519) and then removed using UCHIME [27]. Sequences were assigned to operational taxonomic units (OTUs) at a 97% identity threshold using UPARSE (v7.0.1001) [28]. Taxonomic classifications were assigned using the SILVA SSURef database release 123 and the mothur-based implementation of the RDP classifier [29,30]. Alpha diversity as indicated by the Chao1 and Shannon indices, were analyzed with QIIME and displayed using the ggplot2 package in R [31]. Principal coordinate analysis (PCoA) based on the unweighted UniFrac distance was calculated using QIIME.

### 2.3. Statistical Analysis

The data for growth, blood indices, stomach mass, and rumen fermentation were analyzed using the general linear model (GLM) procedure of SAS (SAS Version 8.01, SAS Institute, Inc., Cary, NC, USA). Duncan’s multiple range tests were conducted when a significant difference was detected among means. Chao1 and Shannon indices were analyzed by *t*-tests to determine whether differences existed within the community diversity. An analysis of similarity randomization test (ANOSIM) [32] was used to calculate the *P*-value and to determine whether differences existed in the microbial composition between the groups. Different bacterial taxa at the genus level were determined with a *t*-test, and only those with a relative abundance >0.1% in at least one sample were visualized using the perl-SVG module (Scalable Vector Graphics, v5.18.2). A value of *p* < 0.05 reported statistical significance. Non-parametric Spearman rank correlation coefficient analysis was conducted using the PROC CORR procedure of SAS (SAS Version 8.01, SAS Institute, Inc., Cary, NC, USA) to detect possible relationships between rumen fermentation parameters and the bacterial community. The threshold of statistically significance at *p* < 0.05 was described to illustrate the relationships. The correlation matrix was visualized using the corrplot package in R [33]. 

## 3. Results 

### 3.1. Growth, Rumen Fermentation, and Blood Indices in Two-Month-Old Female Calves, and Rumen Organ Development in Male Calves

The body weight and average daily gain (ADG) of female calves fed MR were significantly lower than those for calves fed WM or M at 58 days of age (*p* = 0.0273; *p* = 0.0004; Figure 1A, 1B). Starter intake for the WM group was less than that for the M and MR groups (*p* = 0.0249; Figure 1C). The concentration of total volatile fatty acids (TVFA) was greater for the MR group than for the other groups (*p* = 0.0215). The molar proportion of isovalerate was greater for the WM group than for the other groups (*p* = 0.0256; Table 4). The NEFA concentration was lower for the WM group than for the other groups (*p* = 0.0037). The concentrations of GH, h-EGF, and IGF-1, and the GH/insulin ratio were greater for the WM group than for the other groups (*p* = 0.0005; *p* < 0.0001; *p* < 0.0001; *p* < 0.0001, respectively), and the insulin concentration was lower for the WM group than for the other groups (*p* = 0.0141; Table 5). Additionally, the stomach mass as a percentage of the live weight of male calves was lower for the WM group than for the M and MR groups (*p* = 0.0395). The wet mass of rumen expressed as a percentage of the total weight of the four stomachs was significantly lower (*p* = 0.0065), whereas that of the abomasum was significantly greater (*p* = 0.0056) for the WM group than for the other groups (Table 6). 

### 3.2. Rumen Fermentation and Blood Indices in Six-Month-Old Female Calves

No difference was observed in the body weight of calves at 90, 120, 150 and 180 days of age. The ADG of calves during 58 to 180 days of age was similar among groups (Figure 2). The molar proportion of ruminal propionate in the MR group was lower than in the WM and M groups (*p* = 0.0221), whereas the ruminal pH and acetate/propionate ratio in the MR group were significantly greater than in the other groups at six months of age (*p* = 0.0038; *p* = 0.0055, respectively; Table 4). No difference was observed in the blood indices, except for the SUN concentration, which was lower in the MR group (*p* = 0.0303; Table 5). 

### 3.3. Ruminal Bacterial Community

A total of 2,504,870 sequences were generated from 47 samples, with an average of 51,661 retained sequences following quality filtering and chimera removal for each sample. The average length of the retained sequences was 253 base pairs. The overall number of operational taxonomic units (OTUs) detected was 2337 based on a 97% nucleotide sequence identity between reads (Appendix A). The rarefaction curves indicated that the number of each sequence approached a saturation plateau, which indicated that sufficient coverage of all OTUs had been obtained so as to accurately describe the bacterial diversity (Appendix A). *Bacteroidetes* was the most predominant phylum in all samples (65.61%), followed by Firmicutes (20.6%), Proteobacteria (9.99%), Tenericutes (1.20%), Spirochaetes (1.09%), Cyanobacteria (0.44%), Actinobacteria (0.32%), Synergistetes (0.15%), Euryarchaeota (0.08%), Fibrobacteres (0.11%) and unclassified others (0.43%) (Figure 3). The richness of rumen microbiota was greater in the WM2 group compared to that in the M2 group, as indicated by Chao 1 (Figure 4A; *p* = 0.0391). No significant difference was observed in Shannon diversity (Figure 4B).

The principal coordinate analysis (PCoA) that used unweighted Unifrac distances indicated a separation between two-month-old calves and six-month-old calves. The bacterial community structure of WM2 was distinct from M2 (R = 0.2874, *p* = 0.001, ANOSIM) and MR2 (R = 0.1895, *p* = 0.018, ANOSIM), and that of MR 6 was distinct from M6 (R = 0.4955, *p* = 0.003, ANOSIM) and WM6 (R = 0.4542, *p* = 0.005, ANOSIM) (Figure 5). 

The genera Rikenellaceae RC9 gut group, *Butyrivibrio* 2, and *Prevotellaceae* UCG-003 were more abundant, whereas the genus *Prevotella* 7 was less abundant in the WM2 group than in the M2 or MR2 group, (Figure 6A, B). The Rikenellaceae RC9 gut group, *Selenomonas* 1, *Prevotellaceae* UCG-003, Lachnospiraceae NK3A20_group and Ruminococcaceae NK4A214 group were more abundant in the MR6 group, whereas *Prevotella* 7 and *Succinvibrionaceae* UCG-001 were more abundant in the WM6 group (Figure 6C). The Rikenellaceae RC9 gut group, *Prevotellaceae* UCG-003, *Prevotellaceae* UCG-001 and Ruminococcaceae NK4A214 group were more abundant in the MR6 group, whereas *Prevotella* 7 and *Succinvibrionaceae* UCG-001 were more abundant in the M6 group (Figure 6D). No difference was observed between M2 and MR2, M6, and WM6.

### 3.4. Correlation Analysis

Correlations between rumen fermentation parameters and bacterial species are presented in Figure 7. Even though less bacterial species were correlated with the pH, TVFA, acetate, propionate, butyrate, valerate and A/P ratio in two-month-old calves than those in six-month-old calves, more bacterial species were correlated with the NH_3_-N, isobutyrate, and isovalerate in two-month-old calves. Among them, a positive correlation was found for the isovalerate concentration and the abundance of *rumen bacterium* NK4A214, *Lachnospiraceae bacterium* NK4A179, *Butyrivibrio fibrisolvens*, r*umen bacterium* NK4A237, *Treponema bryantii*, SR1 *bacterium canine oral taxon* 369, *rumen bacterium* RC-2, *rumen bacterium NK3B31*, *rumen bacterium* NK4B65, *rumen bacterium* YS3 and *Acidaminococcus fermentans* in two-month-old calves. The abundance of *Prevotella sp.* DJF CP65, *Lachnospiraceae bacterium* DJF B223, *Treponema berlinense, Selenomonas bovis* and *Bacteroides pyogenes* was positively correlated with the propionate concentration, and negatively correlated with the acetate/propionate ratio in six-month-old calves. 

## 4. Discussion 

In calves that have not been weaned, most of their liquid feed intake bypasses the rumen and enters the abomasum directly since the esophageal groove is closed. Abe (1979) [34] reported that the reflex closure of the esophageal groove occurs efficiently and independently of the feeding method (nipple-feeding or bucket-feeding) when calves are familiar with either method. Wise (1984) [35], however, indicated that the reticular groove reflex is more efficient when calves suck nipples rather than drink from buckets. Changes in the feeding method from a floating nipple to bucket have been shown to successfully improve the plasma metabolic and endocrine profiles of ruminal drinking calves [36]. More recent research found that a considerable amount of liquid feed may leak into the rumen, an amount that has been estimated to be 17%–35% of the total intake in bucket-fed calves [37,38,39], as opposed to 0%–20% in nipple-fed calves [40,41]. The bucket feeding method has been widely used because it is labor-saving and easy to perform. This method, however, increase the risk of rumen drinking, which may lead to ruminal fermentation disorders and metabolic acidosis [42]. Subacute ruminal acidosis (SARA) is characterized by a sustained depression in ruminal pH below a value of 5.6 [43] or 5.8 [44] in dairy cows. However, the rumen pH of dairy calves is lower than that of mature cows, averaging between 5.09 and 5.31 [37] and 5.19 and 6.16 [45]. This is primarily because maximum dry feed intake is generally encouraged in preweaning calves in order to promote rumen development. In addition, the leakage of liquid feed into the rumen may result in the bacterial fermentation of milk and lead to ruminal acidosis. In this trial, the ruminal pH was within the normal pH range for calves, possibly suggesting that no abnormal rumen function occurred in the experimental calves.

Waste milk comprises colostrum, milk obtained from mastitic cows, and milk from cows treated with antibiotics. The content of milk protein that entered the rumen might have been greater in the WM group than in the M group. Branched-chain VFA (BCVFA) are synthesized by microorganisms in the rumen via oxidative deamination and the decarboxylation of branched-chain amino acids [46]. Isobutyrate, isovalerate, and 2-methylbutrate are generated from valine, leucine and isoleucine, respectively. The relatively greater concentration of isovalerate in the rumen of the WM group may be associated with the greater protein content of WM. Supplementation of isovalerate increases the population of *Butyrivibrio fibrisolvens* in calf and steer feed [47,48]. This was confirmed by our finding that *Butyrivibrio fibrisolvens* was positively correlated with the concentration of isovalerate in two-month-old calves. Additionally, the populations of *Ruminococcus albus*, *Ruminococcus flavefaciens* and *Fibrobacter succinogenes* have been found to linearly increase with increasing isobutyrate or isovalerate supplementation [47,48,49]. No differences in the abundance of these bacteria were detected in this trial. The increases in these bacterial populations occur mainly because these bacteria firmly adhere to plant tissues when initiating cellulose degradation [50]. In this study, however, we sampled ruminal liquids, which did not include the bacterial populations that attach to feed particles. Deng (2017) [51] reported that the number of OTUs in the rumen digesta of calves fed pasteurized waste milk was higher than that in calves fed untreated whole milk. We observed that, compared to untreated whole milk, feeding pasteurized waste milk increased the ruminal bacterial richness. This may be associated with the concentration of isovalerate in the rumen, as a greater population of total bacteria has been detected in the rumen of steers fed isovalerate [47]. 

The extraruminal effects of BCVFA have been detected by feeding BCVFA to dairy cows: the level of GH increases and that of insulin and NEFA decreased [52,53,54]. Serum GH and IGF-1 for both pre- and post-weaning calves increased linearly with increasing isovalerate supplements [55]. A similar result was observed in our trial: a greater concentration of ruminal isovalerate may induce increases in serum GH and IGF-1 but decreases in insulin and NEFA in the WM2 group. BCVFA receptors are believed to be present in ruminal and hepatic membranes and might perturb the function of hormone-regulated systems, such as those involving insulin and IGF-1 [56,57]. Additionally, bovine colostrum is characterized by high levels of IGF-1 and EGF, which have been found to be resistant to pasteurization [58]. Therefore, the bioavailability of these growth factors in colostrum may contribute to greater concentrations of serum IGF-1 and h-EGF in calves fed WM. Rauprich et al. (2000) [59] reported that calves fed colostrum had greater plasma IGF-1 levels than those fed MR. Insulin, IGF-1, and EGF have been implicated as possible mediators of rumen epithelial cell proliferation and thus play an important role in accelerating rumen development in calves [60,61].

Antibiotic susceptibilities of ruminal bacteria have been previously determined [62]. Chlortetracycline, oxytetracycline, tylosin, and monensin exhibit a strong inhibitory effect on in vitro cellulose digestibility and VFA production in mixed-rumen cultures [63]. Monensin and virginiamycin can alter in vivo rumen microbial populations [64,65]. Feeding calves milk with very low concentrations of ampicillin, ceftiofur, penicillin and oxytetracycline affects the composition of the microbial population in feces [66]. The concentration of antibiotic residues in waste milk cannot be reduced by pasteurization [67]. Therefore, it is likely that a substantial number of bacteria may be sensitive to the presence of antibiotic residuals in WM, which may lead to a distinctive rumen bacterial community of calves in the WM2 group. Gentamicin was the only antibiotic residue detected in waste milk: its concentration was 0.067 ± 0.042 mg/L (mean ± SD). The decreased relative abundance of *Prevotella 7* in the WM2 group might be explained by its susceptibility to gentamicin, a hypothesis supported by a recent study reporting that gentamicin inhibited 90% of *Prevotella intermedia* [68]. A decreased abundance of the genus *Prevotella* was also reported in calves fed waste milk containing 0.024 mg/L penicillin, 0.025 mg/L streptomycin, 0.10 mg/L tetracycline, and 0.33 mg/L ceftiofur [69].

Feeding calves a limited amount of liquid nutrition to encourage rumen development is an effective strategy that promotes the transition from a non-ruminant to ruminant state. The nutrient value of MR was lower than that of WM and M. The digestibility of milk protein ranged from 90% to 97% [70], whereas that of MR containing soy protein ranged from 70% to 78% [4]. Therefore, the amount of digestible protein in calves fed MR may be lower than those fed M or WM, which would account for the slower growth rate. Starter feed intake is a good indicator of rumen development. Morrill (2012) [71] reported that starter feed intake of 700 to 1000 g/d for 3 consecutive days was an adequate weaning criterion. In this trial, we observed that calves in WM consumed less starter feed than the above weaning requirement, which was in accordance with the data of stomach weight obtained from male calves. The starter feed intake was similar for calves in the M and MR groups, although a much greater TVFA level was observed in the MR group. This may suggest poor-efficient VFA absorption in the MR group. The soy protein in the MR would impair intestinal epithelial development, primarily due to the existence of antinutritional factors [3] and increased intestinal abnormalities [72]. As mentioned above, some of the MR may directly leak into the rumen, thus slowing rumen papillae growth. Moreover, MR containing soy protein negatively affects calf growth, metabolic status, and small intestine development, which are effects that may inhibit rumen development indirectly [10]. Starter feed intake is generally considered to be the main stimulator of rumen development. Our results, however, suggest that rumen development is also closely associated with the type and composition of liquid feed.

The postnatal period may be the most critical window for rumen manipulation, and the early feeding regime may lead to permanent changes in the rumen microbial composition [11,12,13]. It has been reported that the structure of the bacterial community established in lambs was affected by the diet fed around the time of weaning; an effect that persists for over four months [14]. Ruminal bacterial communities of lambs can be modified by the diets of the maternal ewes and lambs or by inoculation treatment: this modification lasts until five months of age [15]. We noted a long-lasting effect on rumen fermentation and bacterial community at the age of six months for the calves fed MR. The greater pH and ratio of acetate to propionate in the MR6 group might have a close relationship with the rumen bacterial composition. The species *Prevotella sp.* DJF CP65 is a member of the genera *Prevotella* 7, and its presence positively correlated with the molar proportion of propionate and negatively correlated with the pH value, the molar proportion of acetate, and the acetate/propionate ratio. As the abundance of *Prevotella* 7 was significantly lower in the rumen of calves fed MR, it might explain the differences found for the rumen fermentation parameters. Conversely, a significant increase in the presence of the species *rumen bacterium NK4A214* might contribute to the increased pH, acetate/propionate, and decreased propionate. Reddy (2017) [73] indicated that the early diet affects rumen papillae development, which may result in the differences observed in later carcass traits of beef. Therefore, we speculated that the ruminal imprinting effect in calves fed an MR diet might be associated with the development of the rumen epithelium. Moreover, although the feeding, management, and field conditions for all calves were the same from the ages of two to six months, various barn effects may influence the rumen environment of calves, thus impairing the imprinting evidence from the early feeding regime in this trial. Not all nutritional interventions in the early life of a calf can promote the establishment of different microbial populations in the rumen of the young animal. Providing supplemental plant extracts to calves did not yield long-term effects on rumen fermentation patterns or the bacterial community [16]. Similarly, in our results, we observed that differences in ruminal fermentation and the bacterial community diminished at six months of age. 

## 5. Conclusions 

The early feeding regime impacts rumen development not only by dry matter intake, but also the type of liquid feed. Calves fed waste milk showed a distinct bacterial community structure at two months of age, although this difference diminished by six months of age. Calves fed milk replacer had a greater concentration of total volatile fatty acids at two months of age, which may induce a long-lasting effect on the rumen environment. The early feeding regime may lead to permanent changes in the rumen microbial composition and environment.

## Figures and Tables

**Figure 1 animals-09-00443-f001:**
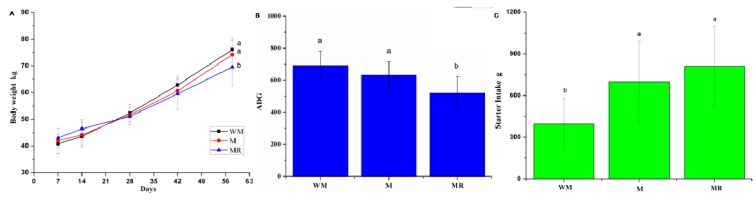
Growth performance of pre-weaning calves (female, *n* = 15 per each treatment). (**A**) Body weight of calves during 7 to 58 days of age; (**B**) average daily gain (ADG) of calves during 7 to 58 days of age; (**C**) starter intake of calves during 54 to 60 days of age; data are expressed as means ± standard deviation. ^ab^ Mean values with different superscripts are different at *p* < 0.05 according to Duncan’s multiple-range test.

**Figure 2 animals-09-00443-f002:**
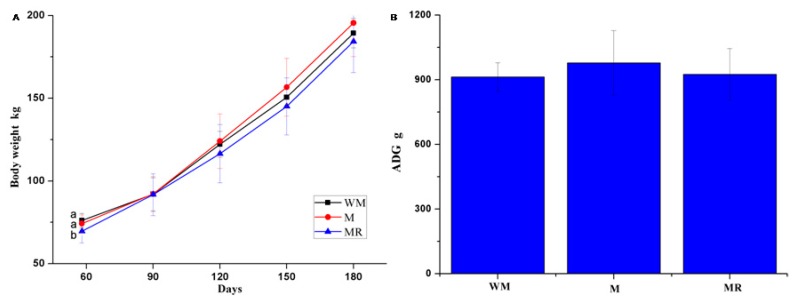
Growth performance of post-weaning calves (female, *n* = 15 per each treatment). (**A**) Body weight of calves during 58 to 180 days of age; (**B**) average daily gain (ADG) of calves during 58 to 180 days of age; data are expressed as means ± standard deviation. ^ab^ Mean values with different superscripts are different at *p* < 0.05 according to Duncan’s multiple-range test.

**Figure 3 animals-09-00443-f003:**
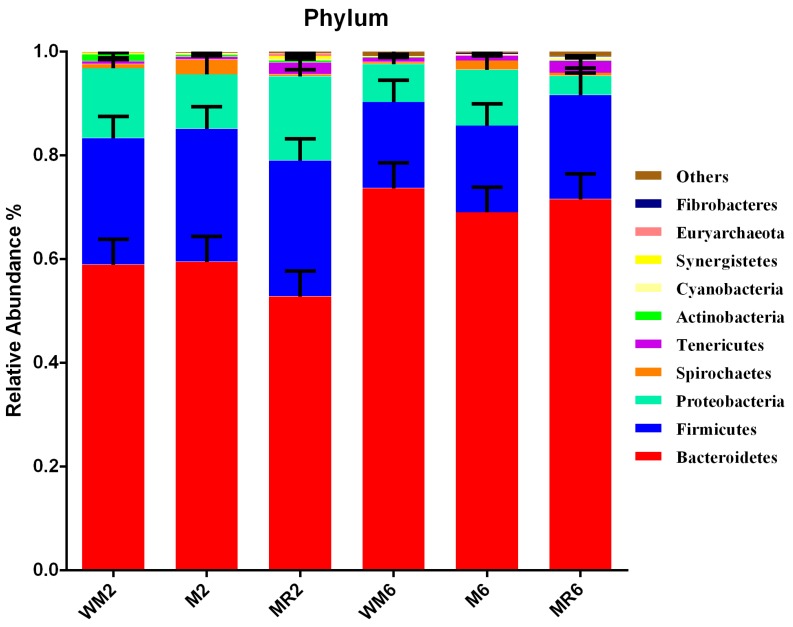
Relative abundance of rumen bacteria at the phylum level. The 10 most abundant phyla are presented. Each bar represents the average relative abundance of each bacterial taxon within a treatment group. Data are expressed as means ± standard error of the mean (SEM). WM2, M2, and MR2: rumen liquid was sampled at the age of two months from calves fed waste milk, milk, and milk replacer, respectively. WM6, M6, and MR6: rumen liquid was sampled at the age of six months from calves fed waste milk, milk, and milk replacer, respectively.

**Figure 4 animals-09-00443-f004:**
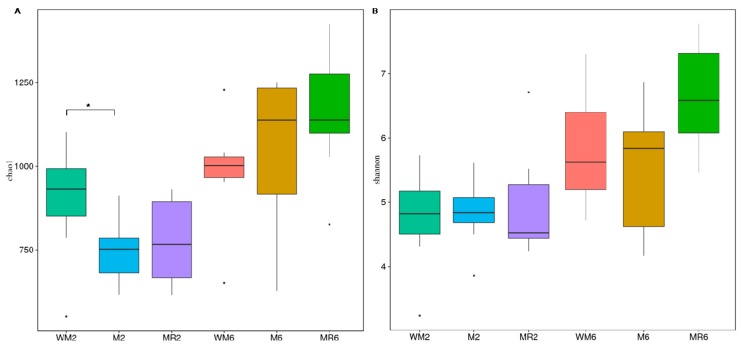
Alpha-diversity measures indicated by the (**A**) Chao1 richness estimator and (**B**) Shannon diversity. The horizontal lines within boxes represent the median, and the tops and bottoms of the boxes represent 75th and 25th quartiles, respectively. Outliers are plotted as individual points. *Significant difference was detected between treatment groups at *p* < 0.05 according to a *t*-test. WM2, M2 and MR2: rumen liquid was sampled at the age of two months from calves fed waste milk, milk and milk replacer, respectively. WM6, M6, and MR6: rumen liquid was sampled at the age of six months from calves fed waste milk, milk, and milk replacer, respectively.

**Figure 5 animals-09-00443-f005:**
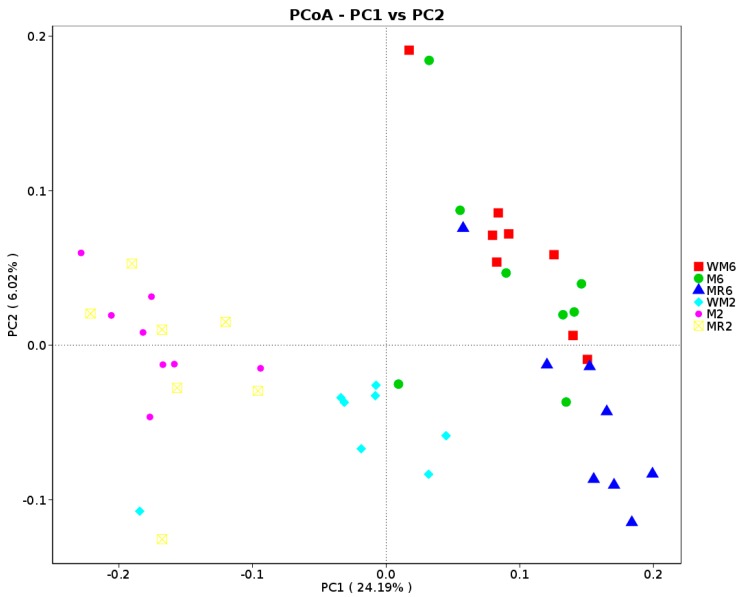
Principal coordinate analysis of beta-diversity of the rumen bacterial community based on unweighted UniFrac distances.. WM2, M2, and MR2: rumen liquid was sampled at the age of two months from calves fed waste milk, milk or milk replacer, respectively. WM6, M6, and MR6: rumen liquid was sampled at the age of six months from calves fed waste milk, milk, or milk replacer, respectively.

**Figure 6 animals-09-00443-f006:**
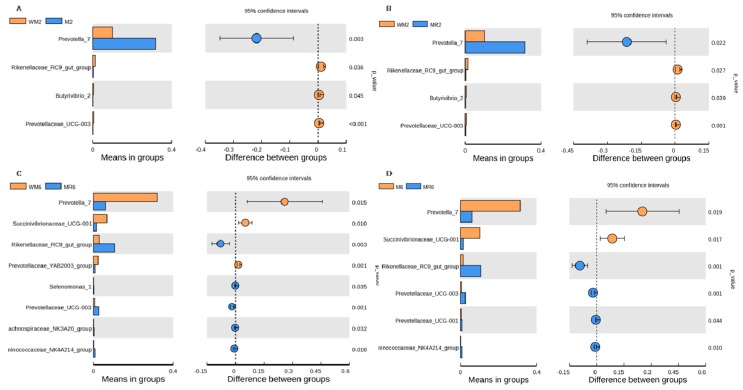
Taxonomic comparisons of the bacterial community at the genus level. Only different bacterial taxa (*p* < 0.05, *t* test) with a relative abundance >0.1% in at least one sample were presented. (**A**) The WM2 and M2 groups; (**B**) the WM2 and MR2 groups; (**C**) the WM6 and MR6 groups; (**D**) the M6 and MR6 groups. WM2, M2, and MR2: rumen liquid was sampled at the age of two months from calves fed waste milk, milk, or milk replacer, respectively. WM6, M6, and MR6: rumen liquid was sampled at the age of six months from calves fed waste milk, milk, or milk replacer, respectively.

**Figure 7 animals-09-00443-f007:**
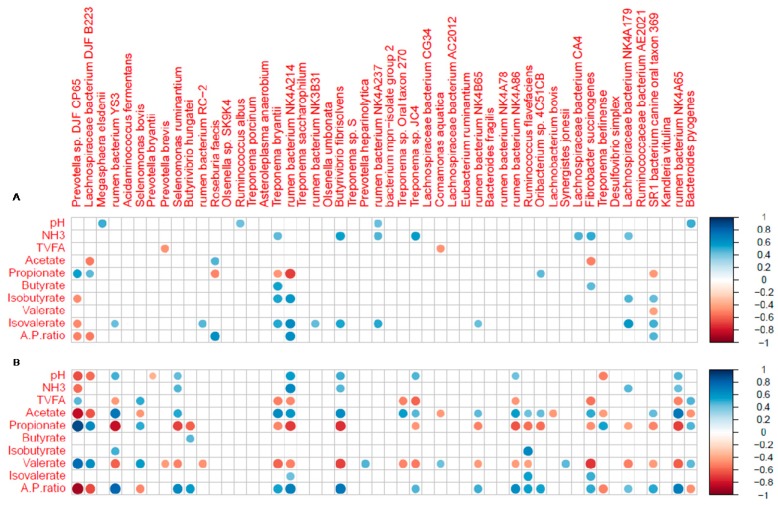
Spearman’s rank correlations between rumen fermentation parameters and relative taxa abundance in (**A**) two-month-old and (**B**) six-month-old calves. Only bacterial species with a relative abundance >0.01% in at least one sample were included in the correlation analysis. Correlations with a threshold of statistical significance at *p* <0.05 were visualized. The degree of correlation is indicated by the size of the circle. The blue color represents a positive correlation and the red color represents a negative correlation.

**Table 1 animals-09-00443-t001:** The amount of liquid feed during 7 to 63 days old.

Age	Feeding Amount (L)
7–14 d	5.0
14–28 d	5.4
28–42 d	6.2
42–60 d	7.3
61 d	5.5
62 d	3.7
63 d	1.8

**Table 2 animals-09-00443-t002:** Chemical analysis of the milk, waste milk and milk replacer emulsion.

Chemical Analysis	Milk ^1^	Waste Milk ^1^	Milk Replacer Emulsion ^2^
Crude Protein (%)	3.3 ± 0.27	4.29 ± 0.61	2.71 ± 0.01
Crude Fat (%)	4.25 ± 0.25	3.91 ± 0.56	1.90 ± 0.04
Crude Lactose (%)	4.47 ± 0.27	3.85 ± 0.52	-

^1^ Milk and pasteurized waste milk (WM) were sampled biweekly. ^2^ Milk replacer (MR) (patent number, CN02128844.5) was purchased from the Beijing Precision Animal Nutrition Center, Beijing, China. Data were calculated based on a chemical analysis of the milk replacer powder. The crude protein, fat, netural detergent fiber, acid detergent fiber, ash, calcium, and total phosphate of milk replacer powder on a dry matter basis are 22.93%, 16.02%, 5.07%, 1.52%, 4.3%, 0.9%, and 0.49%, respectively. The ingredient compositions are soy flour, whole milk powder, whey permeate, starch dextrin, calcium carbonate, dicalcium phosphate, lysine, methionine, threonine, vitamin premix, trace minerals premix, and additives.

**Table 3 animals-09-00443-t003:** Composition and chemical analysis of the concentrate feeds and forage (dry matter basis).

Ingredients	Concentrate Feed		Forage ^2^	
	0–3 Months	4–6 Months	Alfalfa Hay	Oat Hay
Corn (%)	55.65	56.5		
Soybean Meal (%)	26.2	23.15		
Extruded Soy (%)	7	0		
Wheat bran (%)	3.9	10		
Distillers Dried Grains with Solubles (%)	3	6		
Calcium Carbonate (%)	2.25	2.302		
Phospate Dicalcium (%)	0.6	0.55		
Salt (%)	0.4	0.5		
Premix (%) ^1^	1	1		
Items	0–3 Months	4–6 Months	Alfalfa hay	Oat hay
Dry Matter (%)	87.94	87.77	90.22	89.50
Crude Protein (%)	20.00	18.00	18.77	7.18
Crude Fat (%)	3.86	3.21	2.32	2.27
Neutral Detergent Fiber (%)	9.79	11.41	35.31	50.33
Acid Detergent Fiber (%)	3.77	3.95	26.22	32.36
Crude Ash (%)	6.94	6.99	7.35	5.57
Calcium (%)	1.00	1.00	1.28	0.32
Total Phosphate (%)	0.45	0.45	0.26	0.20
Salt (%)	0.47	0.58	-	-

^1^ provided per kg of basal diet: 10 000 IU vitamin A, 1 500 IU vitamin D, 60 IU vitamin E, 1.5 mg vitamin B1, 8.2 mg vitamin B2, 2.0 mg vitamin B6, 3.6 mg vitamin K, 1.0 mg folic acid, 0.1 mg biotin, 49.5 mg niacin, 60.0 mg D-pantothenic acid, 10.2 mg Cu, 20 mg Fe, 140 mg Zn, 140 mg Mn, 2.0 mg I, and 0.44 mg Se. ^2^ The composition of forage is 50% alfalfa hay and 50% oat hay during 2–3 months, and 30% alfalfa hay and 70% oat hay during 4–6 months.

**Table 4 animals-09-00443-t004:** Rumen fermentation parameters of calves at two and six months of age (female, *n* = 8 per treatment).

Items	Age	Treatment ^3^			SEM	*p* value
		WM	M	MR		
pH	2-month-old	6.28	6.15	5.67	0.174	0.0651
	6-month-old	6.38 ^b^	6.36 ^b^	6.71 ^a^	0.077	0.0038
NH_3_-N(mg/dL)	2-month-old	30.08	20.15	19.17	4.337	0.1787
	6-month-old	9.67	10.22	11.92	1.063	0.3484
TVFA (mmol/L) ^1^	2-month-old	39.21 ^b^	40.45 ^b^	62.13 ^a^	5.269	0.0215
	6-month-old	71.82	70.29	61.5	4.888	0.3263
Acetate (%) ^2^	2-month-old	49.48	47.15	44.39	2.693	0.4483
	6-month-old	60.92	61.86	66.72	1.698	0.0683
Propionate (%) ^2^	2-month-old	31.03	35.29	38.87	2.39	0.1051
	6-month-old	25.52 ^a^	25.49 ^a^	19.14 ^b^	1.63	0.0221
Butyrate (%) ^2^	2-month-old	11.95	10.59	11.49	1.863	0.8731
	6-month-old	9.46	8.76	10.18	0.759	0.4580
Isobutyrate (%) ^2^	2-month-old	1.57	0.91	0.79	0.314	0.2055
	6-month-old	0.90	0.83	1.08	0.129	0.3980
Valerate (%) ^2^	2-month-old	2.94	4.37	3.02	0.608	0.2041
	6-month-old	1.57	1.66	1.21	0.125	0.0565
Isovalerate (%) ^2^	2-month-old	3.03 ^a^	1.69 ^b^	1.45 ^b^	0.399	0.0256
	6-month-old	1.63	1.41	1.67	0.148	0.4450
Acetate/propionate	2-month-old	1.77	1.36	1.2	0.204	0.1624
	6-month-old	2.44 ^b^	2.55 ^b^	3.70^a^	0.255	0.0055

^1^ TVFA: total violate fatty acids. ^2^ Acetate (%), propionate (%), butyrate (%), isobutyrate (%), valerate (%), and isovalerate (%) imply the molar proportion of each to that of the TVFA. ^3^ WM: waste milk; M: whole milk; MR: milk replacer. ^ab^ Mean values within a row with different superscripts differ.

**Table 5 animals-09-00443-t005:** Blood metabolites and hormones concentrations of calves at two and six months of age (female, *n* = 10 per each treatment).

Items	Age	Treatment ^7^			SEM	*p*-value
		WM	M	MR		
SUN (mmol/L) ^1^	2-month-old	5.64	5.65	5.03	0.5232	0.6866
	6-month-old	8.09 ^a^	8.17 ^a^	6.78 ^b^	0.3896	0.0303
NEFA (mmol/L) ^2^	2-month-old	0.37 ^b^	0.49 ^a^	0.44 ^a^	0.0232	0.0037
	6-month-old	0.41	0.42	0.41	0.0162	0.8815
GH (ng/mL) ^3^	2-month-old	4.70 ^a^	3.93 ^b^	3.94 ^b^	0.1385	0.0005
	6-month-old	4.58	4.10	3.98	0.2647	0.2449
Insulin (IU/mL)	2-month-old	9.17 ^b^	18.53 ^a^	15.29 ^ab^	2.1231	0.0141
	6-month-old	14.53	14.95	13.82	1.8822	0.9128
GH/insulin ^4^	2-month-old	0.63 ^a^	0.26 ^b^	0.27 ^b^	0.0482	<0.0001
	6-month-old	0.36	0.30	0.33	0.0447	0.6181
h-EGF (ng/mL) ^5^	2-month-old	0.95 ^a^	0.82 ^b^	0.81 ^b^	0.0181	<0.0001
	6-month-old	0.93	0.76	0.87	0.083	0.3506
IGF-1 (ng/mL) ^6^	2-month-old	219.31 ^a^	168.46 ^b^	167.44 ^b^	7.1208	<0.0001
	6-month-old	186.54	177.06	195.43	19.245	0.7977

^1^ SUN: serum urea nitrogen; ^2^ NEFA: non-esterified fatty acid; ^3^ GH: growth hormone; ^4^ GH/insulin: growth hormone/insulin ratio; ^5^ h-EGF: human epidermal growth factor; ^6^ IGF-1: insulin-like growth factor; ^7^ WM: waste milk; M: whole milk; MR: milk replacer; ^ab^ mean values within a row with different superscripts differ.

**Table 6 animals-09-00443-t006:** Stomach development for two-month-old calves (male, *n* = 3 per each treatment).

Items	Treatments ^3^			SEM	*p* value
	WM	M	MR		
Stomach mass ^1^	1.26 ^b^	2.12 ^a^	2.19 ^a^	0.21	0.0395
Rumen % ^2^	42.05 ^b^	51.89 ^a^	55.05 ^a^	1.88	0.0065
Reticulum % ^2^	7.75	8.53	8.83	0.50	0.3603
Omasum % ^2^	13.28	11.13	10.55	0.93	0.1746
Abomasum % ^2^	36.93 ^a^	28.45 ^b^	25.58 ^b^	1.58	0.0056

^1^ Stomach mass is expressed as a percentage of the live weight of calves. ^2^ Individual mass of each stomach compartment as a percentage of the total weight of the four stomachs.^3^ WM: waste milk; M: whole milk; MR: milk replacer. ^ab^ Mean values within a row with different superscripts differ.

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
