# Peer review of "Early Feeding Regime of Waste Milk, Milk, and Milk Replacer for Calves Has Different Effects on Rumen Fermentation and the Bacterial Community"

_animals, 2019, doi:10.3390/ani9070443_

Round 1
Reviewer 1 Report
This is overall a good manuscript, which describes interesting and relevant data. I have some recommendations for revision, before I would recommend publishing:
In general: Please carefully check spelling, and use of commas (e.g., use the Oxford comma continously) to improve readability and understandability of the text.
Introduction: L62: Which inoculation treatment? Please clarify.
L65: Which plant extracts? Please give examples.
Materials and methods: Please sort the sentences on 2.1. in a meaningful manner, and please delete repetitions. Use past tense continously (also in the rest of the text).
L76: 5 L per day? In table S1 you showed different quantities of liquid feed dependent on age. Insert table S1 in the text, I think it is important.
Please insert tables S2 and S3 in the main text as well. Include MR composition and nutrient composition in the text.
L84: Were the calves (15 per barn) randomly assigned? Add a statement.
L95-97: Why did you only measure intake of the concentrate, and not that of forage, only in the selected females, and only during these 7 days?
L103-107: Should better appear in the discussion.
L107-114: Please give adequate references, or more detail for the analyses.
Why did you use only the females for blood metabolite measurements, but only males for assessment of organe development? Please explain. I think it would be better to do both with both sexes.
2.2.4.: Give more details on the analytical methods.
L138: 16srRNA not rDNA.
Results: L241-242: "... was positively correlated" is written italic.
Discussion: L319: Did you measure digestible protein? Show the data.
L326-327: Please better explain, why soy protein impair epithelial development.
Conclusion: It is too superficial. Please add a statement that highlights the implications of your work.
References: Please carefully check for correctness here, e.g., use journal title abbreviations continously.
Tables 3 and 4: Could be combined with tables 1 and 2, respectively.
Fig. 1: Please define ADG.
Table S2: Delete information on antibiotics, and add this to the text.
Table S3: I miss analyses on forage, and information, what exactly has been fed. Please add, preferably to the text.
Reviewer 2 Report
This is an interesting work evaluation the effect of different liquid feed to dairy calves. Findings are relevant and will contribute to the area of knowledge. However, the manuscript will require to go through an English editing process to improve the quality of the work. Moreover, some sections are not well organized and require re-structuring.
The authors have only selected males for slaughter at the age of 2 months. Is there any reason to believe that the results found in males would differ from those in females?
Some specific comments:
Abstract: the second half of the abstract is not easy to understand (from L33). Please check English and rephrase.
Keywords: do not repeat the same keywords as in the title; please replace “early feeding regime”, “rumen fermentation” and “bacterial community”. If you use different keywords not included in the title that will increase the visibility of your paper (it will be found with a greater number of search criteria).
Introduction:
· (M), (WM) and (MR) were already defined in the abstract section. They should be introduced once.
· The last two sentences of the first paragraph (L53-L55) are disconnected from their precedent text. In general, the flow and connectivity of the Introduction should be improved.
Materials and methods:
· This is section 2. (not 1. As in the manuscript)
· Please write all this section in past tense (e.g. “is” must be replaced by “was”).
· The number of female and male calves per group should be clarified (L72).
· This section is poorly organized, and it is not clear the sequence of feeding regimes. Apparently, after feeding colostrum within the first 24h the calves were trained to eat from the buckets using the three different feeds. After that, apparently a transition MR was supplied. I assume it was different than the used for the experimental period? And the from day 7 they start receiving again the different experimental feeds? Authors describe the weaning process (day 60 to day 63) before describing the pre-weaning period (Day 7 to day 63). Then they mention “15 calves per barn”… I assume those are the female calves only? It was not clarify how 54 calves became 45.
· The sampling described in L85 should be moved to the next section “Sampling and measurements”
· Where is Table S2 and S3? Moreover, they should describe how these nutrient profiles were determined.
· Section 2.2.1. should be “Growth performance and Feed intake”
· The sentences describing the relevance of SUN, NEFA, GH, etc. (L103-106) are not part of M&M. They should be moved to the Discussion.
· Subtitle in L135 should be in italic.
· Statistical analysis (L163) should be section 2.3.
Results: this should be section 3.
· Why are you using findings as a title for this section? The title should be “Growth, Rumen Fermentation, and Blood Indices in 2-Month-Old Female Calves, and Rumen Organ Development in Male Calves”. The same applies for the remaining headings of the Result section
· (L201) “expert”? might be “except”
· Where is Table S3 and S4 (you refer to S1, S2 and S4 only, but none of those tables are available). Where is Fig. S1?
· (L214) WM2 was more diver than M2 only? Or was also more diverse than MR2?
· You present results from two different diversity indices (Chao and Shannon) with contrasting results. Therefore, do the communities differ or not?
· (L221-222) the first sentence of the paragraph should be moved to M&M section
· What do the authors mean by over-represented?
Discussion:
· (L266-267) this sentence contains text that belongs to the M&M section and to the Results sections. Please split and move to the correct sections.
· (L287) “BCFA”
· (L297) this reference is not cited following the journal format [xx]
· (L308-309) How was the antimicrobial content determined? This should be described in the M&M section.
Conclusions: I would spell out the names of the acronyms in this section.
Round 2
Reviewer 1 Report
Well done. Thank you.
Author Response
Thank you!
Reviewer 2 Report
Just minor comments:
L43 – do not start sentences with acronyms. Please spell out the liquid feeds’ name
L60 – De Barbieri citation does not follow journals’ format
Is there any reference for the National Standard in China? If we do not read Chinese we will understand how the samples were analysed.
The authors explain how the Gentamicin was determined but for the other components they only added a footnote in the tables (2 and 3). This is not the usual approach.
Tables – footnotes do not follow journal’s format
Figure 4 – footnote: Chao1 is diversity and Shannon is richness, correct? This should be reflected in the footnote
Author Response
Please see the attachement. Thank you!
